# Study on Restoring Force Performance of Corrosion Damage Steel Frame Beams under Acid Atmosphere

**Bin Wang [1,*], Weizeng Huang [2] and Shansuo Zheng [2]**

[1]  School of Civil and Architecture Engineering, Xi'an Technological University, Xi'an 710021, China
[2]  School of Civil Engineering, Xi'an University of Architecture and Technology, Xi'an 710055, China; huangweizeng@163.com (W.H.); zhengshansuo@263.net (S.Z.)
*  Correspondence: wangbin2346@xatu.edu.cn; Tel.: +86-135-1912-1853

**Abstract:** In order to study the restoring force characteristics of corroded steel frame beams in an acidic atmosphere, based on different corrosion damage degrees, tests on the material properties of 48 steel samples and six steel frame beam specimens with a scale ratio of 1/2 under low cyclic repeated loading were conducted. According to the test results, the relationship between the weight loss rate and the mechanical properties of corrosion damage steel was obtained by numerical regression analysis, and the hysteresis curves and skeleton curves of the corroded steel frame beams were also obtained. The simplified trilinear skeleton curve model of the corroded steel frame beams and the expression of the corresponding feature points were determined by analyzing the failure process. The strength and stiffness degradation rule of the steel frame beam was analyzed furtherly. The hysteresis rule was established by introducing the cyclic degradation index which considers the effect of different corrosion degrees, and finally the restoring force model based on the corroded steel frame beams in an acidic atmospheric environment was established. Comparison with the test results show that the skeleton curve and the restoring force model established in this paper can accurately describe the seismic performance of corrosion damaged steel frame beams and can provide a basis for the seismic calculation analysis of corroded steel structures in an acidic atmosphere.

**Keywords:** steel frame beam; corrosion damage; acidic atmosphere; restoring force

## 1. Introduction

Owing to an excellent seismic-resistant performance, light weight, and being easy to connect, steel beams are widely used as connecting beams or outer frame beams [1–3] in high-rise and super-high-rise hybrid structures. However, there are some defects in steel, such as poor fire resistance and instability, which hold back the practical use of steel members in industry [4–6]. On the other hand, steel can be quickly corroded, especially in extremely harsh environments (e.g., acid rain atmosphere), which will cause the decrease of strength, stiffness, and ductility and affect the seismic-resistant performance of the structure [7,8]. Generally, there are three kinds of protection against corrosion, namely, surface-coating, the cathodic system, and the corrosion resistance of material, which is selected according to the structural type and the surrounding conditions. For instance, Albrecht et al. [9] provided data of thickness loss in structural steel members under various catoptrical conditions and determined the effect of the material properties on the resistance to corrosion. Corrosion has been regarded as one of the five main damages in marine structures [10] and is induced by the thinning of structural material which decreases the strength capacity. A review of the protection against corrosion and the relationship between corrosion and a structural strength capacity can be found in Wang et al. [11], particularly, in the marine environment. At present, there are some studies on the rebars corrosion in reinforced concrete

structures [12,13]. However, few studies on the seismic performance of the damage of corrosion in steel structures and members [14] has been reported.

The corrosion damage of steel can cause the degradation of the structural mechanical properties [15,16]. Therefore, the existing seismic analysis methods for undamaged steel members overestimate the seismic performance of steel members damaged by corrosion, which leads to a significant loss of property [17]. In order to develop a seismic computation analysis of structures during earthquakes, it is necessary to establish an accurate restoring force model of the structure members that is able to reflect the change of structural mechanical properties, such as the strength and stiffness degradation, and the reduction of ductility or the energy-dissipating capacity during an earthquake. The restoring force model consists of the skeleton curve and the hysteresis rule of members [18,19].

The restoring force model can be divided into two categories, namely, a polygonal hysteretic (PH) model and a smooth hysteretic (SH) model, which are very important for the seismic analysis of structures. The most classical SH model is the Bouc-Wen model that considers the relationship between the stiffness, the strength degradation, and the hysteretic energy consumption [20,21]. In the follow-up study, the Bouc-Wen model was improved further by considering the relationship of the stiffness degradation, the maximum displacement, and the effects of the loading path [22,23]. Based on the experimental results of concrete members and the stress softening phenomenon caused by cyclic loading, Wang et al. established an SH model that considers the effect of the reloading path and the cumulative damage [24].

In contrast with the SH model, the PH model has more advantages, such as a simple calculation, more obvious features in each stage, and more consistency with experimental observations. Therefore, the PH model is used more widely in engineering practice. The more classic PH model includes the bilinear and trilinear hysteresis models, which consider several factors, including the cracking of concrete, yielding, loading and unloading stiffness degradation, etc. [25–27]. In this paper, the polyline hysteresis model is adopted for the seismic analysis of corroded steel beams.

The existing results show that there was a relationship between the surface roughness of the steel, the steel thickness, and the hysteretic energy. Meanwhile, steel components with severe corrosion exhibit a phenomenon of early cracking while in use. In addition, the cracking of the steel affects the stiffness degradation of members and the hysteresis performance of components [28,29], which can be obtained by the analysis of the corroded steel members. Although many restoring force models of steel components were obtained by researchers in recent decades [30–33], there is little research on the hysteretic behavior of corroded steel frame beams in a harsh environment and cyclic loading, especially in an acidic atmosphere.

Based on the above-mentioned introduction, the existing PH models fail to fully consider the characteristics of corroded steel beams and cannot accurately reflect the hysteretic behavior of steel beams in an acid rain environment, since the stiffness and strength degradation are obviously caused by corrosion. Therefore, in this paper, based on the experimental study of corroded steel frame beams under different corrosion degrees, the expression of the mechanical characteristics of the steel material was established, and the skeleton curve and hysteresis rule of the corroded steel beam were obtained. Furthermore, the restoring force model of the steel frame beam in an acidic atmosphere was established. The validity of the model was verified by comparing the test results with computational values.

## 2. Experimental Program

The steel material and beam components with different degrees of corrosion were tested through a tensile and low cyclic loading test, respectively, to study the degradation law of mechanical properties of steel material and the degradation law of seismic behavior of steel beams in an acid atmosphere environment.

### 2.1. Steel Material

To obtain the mechanical properties of the corroded steel, such as the yield strength, the ultimate strength, the elongation, and elastic modulus, the tests were conducted based on standards. Then, the above-mentioned fundamental parameters to be used in the seismic analysis of the corroded frame beam were determined in this paper. The thickness of 6.5 mm, 9 mm, and 14 mm of steel plate was constructed according to the Chinese standard "steel and steel products—location and preparation of samples and test pieces for mechanical testing" [34]. The steel grade was Q235. The steel plate samples were corroded through the accelerated corrosion test that simulate an acid atmosphere environment. Each thickness of the steel material samples was divided into 8 sets, including 2 for each set as shown in Figure 1. Then, the steel material samples were placed in the ZHT/W2300 climate simulation experiment box for accelerated corrosion testing in an acidic atmosphere environment. According to the design code of China (GB/T 24195-2009) "Corrosion of metals and alloys—acceleration cyclic test with exposure to acidified salt spray, "dry" and "wet" conditions" [35], the acid salt spray was made, and the preparation process is shown in Figure 2. The specific design parameters of the material test are shown in Table 1.

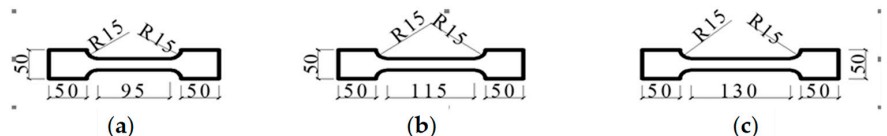

**Figure 1.** Steel material samples (unit: mm). (**a**) Thickness of 6.5 mm; (**b**) thickness of 9 mm; (**c**) thickness of 14 mm.

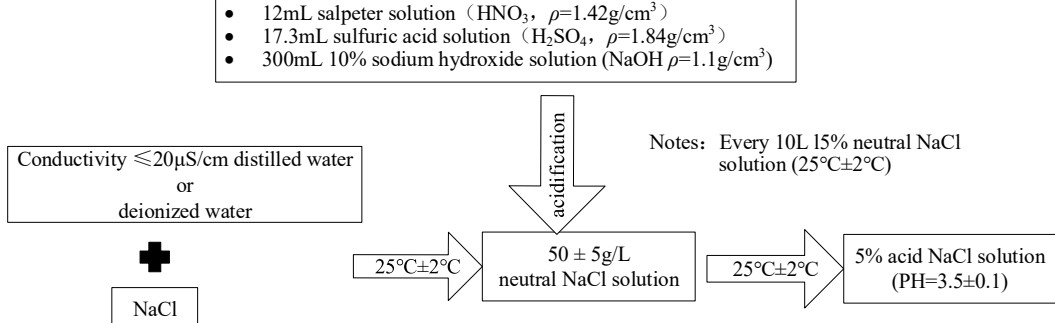

**Figure 2.** The preparation process of acid salt spray.

**Table 1.** Testing parameters of material specimens.

| Specimen Thickness /mm | Quantity | Acceleration Corrosion Time /h |
|---|---|---|
| 6.5 | 16 | |
| 9 | 16 | 0/240/480/960/1440/1920/2400/2880 |
| 14 | 16 | |

After each cycle, which lasts 8 h during the salt spray cycle test, the steel samples were taken out and washed by dilute NaCl solution to remove corroded steel, and then weighed after drying. In order to quantitatively establish the relationship between the mechanical properties of the steel and the degree of corrosion, the weight loss ratio of the sample with different degrees of corrosion was calculated [36]. The specific calculation formula is:

$$D_w = (W_0 - W_1)/W_0 \tag{1}$$

where $D_w$ is the weight loss ratio, and $W_0$, $W_1$ are the weight of the uncorroded and corroded steel material sample, respectively. In this paper, the average value of the weight loss rate of samples of different thickness is adopted to simplify the calculation.

After the test of the salt spray cycle of each steel material sample, the single tensile test was carried out by a universal testing machine to obtain the mechanical properties of the steel. The whole test process of the corroded steel is shown in Figure 3.

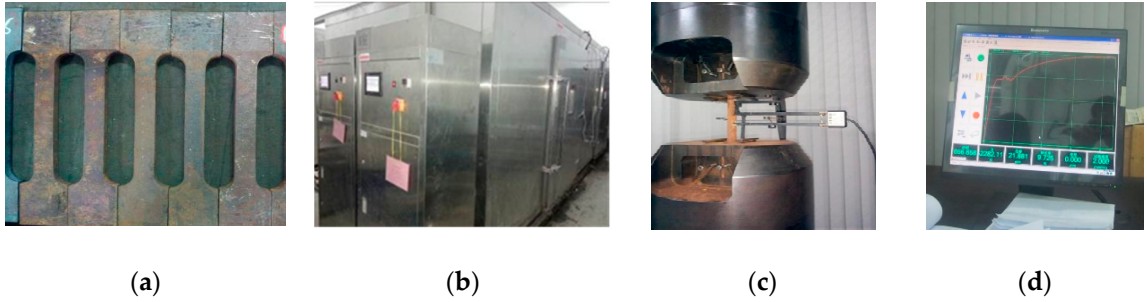

(**a**)　　　　　　(**b**)　　　　　　(**c**)　　　　　　(**d**)

**Figure 3.** Test for the mechanical properties of steel under acidic atmosphere. (**a**) Steel material sample; (**b**) ZHT/W2300 corrosion box; (**c**) tensile test; (**d**) data collection.

## 2.2. Steel Frame Beams

Based on the condition of the loading equipment and the measuring range of the instruments, a total of 6 steel frame beam specimens with a scale of 1:2 were designed by following the current Chinese standard [37–39]. The steel grade is the same as Section 2.1. The specific test parameters are shown in Table 2, and the beams and section dimension are shown in Figure 4.

**Table 2.** Relevant parameters of beam specimens.

| Specimen Number | Sectional Dimension /mm | | Corrosion Time /h | Weight Loss Ratio/% |
|---|---|---|---|---|
| | Beam/mm | Bearing Beam/mm | | |
| B-1 | | | 0 | 0 |
| B-2 | | | 480 | 2.11 |
| B-3 | | | 960 | 4.30 |
| B-4 | HN300 × 150 × 6.5 × 9 | HN350 × 350 × 10 × 14 | 1920 | 7.50 |
| B-5 | | | 2400 | 9.50 |
| B-6 | | | 2880 | 11.28 |

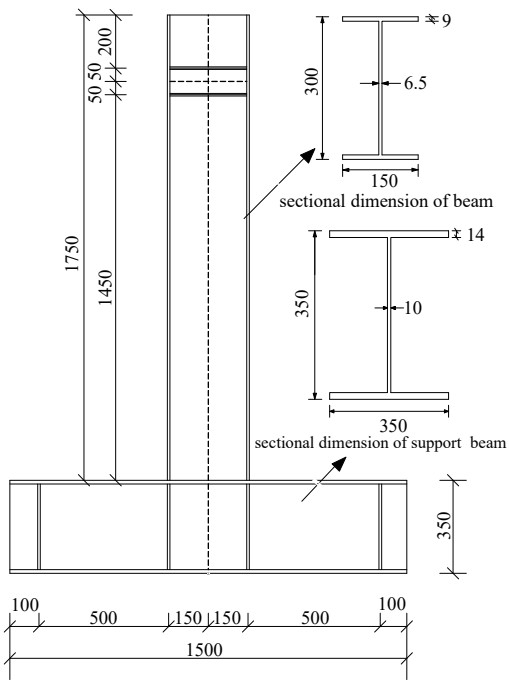

**Figure 4.** Specimen and section dimension (unit: mm).

### 2.3. The Loading Program

A cantilever beam was adopted in loading. The low cyclic loading device is shown in Figure 5. The lateral cyclic loading was applied by a 30 ton MTS actuator, and the base beam was fixed to the rigid ground by the pressure beam and anchor bolts. At the same time, the lateral support was added on both sides to prevent the out-of-plane instability of the beam specimen during loading, though the deflection may have been slight. The horizontal loading scheme was applied by referring to prior investigation [40]. The specimens were subjected to four successive cycles at the displacement drift of 0.375%, 0.5%, 0.75%, 1% respectively, after which followed two successive cycles at the displacement drift of 2.0%, 3.0%, 4%, 5% etc., until the horizontal load dropped to 85% of the peak load of specimens or until the load could not be continued because of the apparent damage, then the loading was stopped. The specific loading scheme is shown in Figure 6.

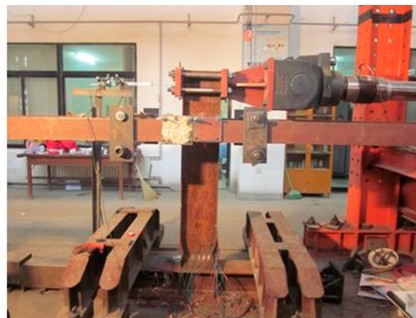

**Figure 5.** Test loading setup.

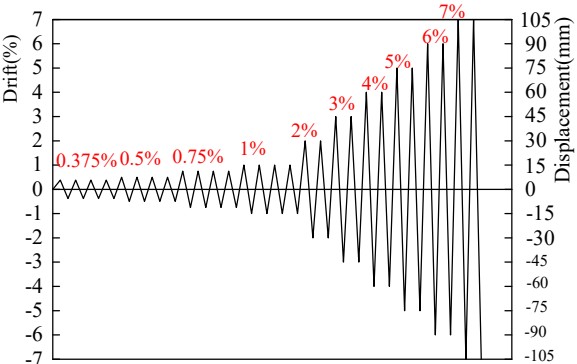

**Figure 6.** Specimen loading scheme.

## 3. Experimental Results

### 3.1. Steel Material

The mechanical properties of the steel obtained by uniaxial tensile tests after different corrosion intervals are shown in Table 3.

**Table 3.** Mechanical properties of steel.

| Thickness (mm) | Corrosion Time (h) | Weight Loss Ratio (%) | Yield Stress $f_y$ (MPa) | Ultimate Stress $f_u$ (MPa) | Elongation $\delta$ (%) | Elasticity Modulus $E_s$ (MPa) |
|---|---|---|---|---|---|---|
| | 0 | 0 | 335.23 | 482.81 | 32.66 | 206,386 |
| | 240 | 1.32 | 330.54 | 486.54 | 32.31 | 206,126 |
| | 480 | 2.54 | 332.86 | 479.55 | 30.14 | 205,003 |
| | 960 | 5.15 | 320.13 | 464.66 | 30.68 | 199,548 |
| 6.5 | 1440 | 6.99 | 309.52 | 456.71 | 29.13 | 198,368 |
| | 1920 | 8.99 | 308.39 | 450.32 | 28.34 | 190,684 |
| | 2400 | 11.35 | 303.17 | 448.65 | 28.07 | 186,684 |
| | 2880 | 13.51 | 291.66 | 441.94 | 24.76 | 180,984 |

**Table 3.** *Cont.*

| Thickness (mm) | Corrosion Time (h) | Weight Loss Ratio (%) | Yield Stress $f_y$ (MPa) | Ultimate Stress $f_u$ (MPa) | Elongation $\delta$ (%) | Elasticity Modulus $E_s$ (MPa) |
|---|---|---|---|---|---|---|
| | 0 | 0 | 341.38 | 493.62 | 30.79 | 205,881 |
| | 240 | 0.93 | 344.23 | 488.64 | 30.52 | 204,111 |
| | 480 | 1.81 | 336.54 | 481.38 | 30.24 | 200,684 |
| 9 | 960 | 3.72 | 330.58 | 478.55 | 29.37 | 199,844 |
| | 1440 | 5.02 | 324.84 | 476.64 | 28.46 | 192,336 |
| | 1920 | 6.49 | 322.16 | 464.22 | 28.10 | 191,558 |
| | 2400 | 8.24 | 322.33 | 467.21 | 26.37 | 188,955 |
| | 2880 | 9.76 | 311.94 | 460.27 | 25.13 | 185,684 |
| | 0 | 0 | 326.64 | 481.58 | 34.17 | 204,768 |
| | 240 | 0.59 | 325.21 | 488.64 | 33.84 | 201,335 |
| | 480 | 1.13 | 322.18 | 479.68 | 33.28 | 202,351 |
| 14 | 960 | 2.39 | 320.44 | 473.26 | 32.82 | 190,667 |
| | 1440 | 3.19 | 315.62 | 474.62 | 32.23 | 197,684 |
| | 1920 | 4.17 | 312.58 | 466.63 | 31.81 | 195,558 |
| | 2400 | 5.26 | 305.11 | 465.21 | 30.67 | 194,668 |
| | 2880 | 6.04 | 302.11 | 459.86 | 28.89 | 191,334 |

As can be seen in the above table, the deterioration of mechanical properties increased with the increase of the steel corrosion time. The longer the corrosion time of the steel, the greater the weight loss ratio. Additionally, the thicker the steel workpiece was, the less the weight loss ratio was. As the weight loss ratio increased, the mechanical properties of steel deteriorated continuously. The relationship of the weight loss ratio and the mechanical properties of the steel was obtained through the linear regression analysis of the above data. The fitting results and the regression expression are shown in Figure 7 and Equation (2), respectively.

$$\begin{cases} f'_y/f_y = 1 - 0.9721D_w \\ f'_u/f_u = 1 - 0.9862D_w \\ \delta'/\delta = 1 - 1.6257D_w \\ E'_s/E_s = 1 - 0.9019D_w \end{cases} \tag{2}$$

where,

$$f'_y = F'_y/A \tag{3}$$

$$f'_u = F'_u/A \tag{4}$$

where $f_y$, $f_u$, $\delta$, $E_s$ are yield strength, unlimited strength, elongation and elasticity modulus of uncorroded steel respectively, $f'_y$, $f'_u$, $\delta'$, $E'_s$ are yield strength, unlimited strength, elongation and elasticity modulus of corroded steel respectively, $F'_y$, $F'_u$ are yield and unlimited tension of corroded steel respectively, and $A$ is section area of uncorroded steel samples.

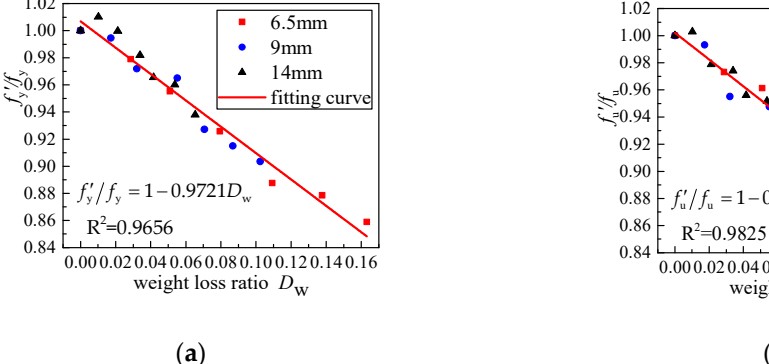

(a)

(b)

**Figure 7.** *Cont.*

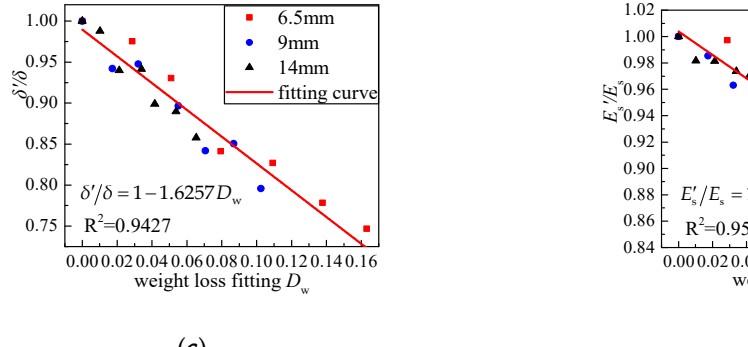

(c)                      (d)

**Figure 7.** Regression fitting results of mechanical properties and weight loss ratio. (**a**) Yield strength; (**b**) ultimate strength; (**c**) elongation; (**d**) elasticity modulus.

## 3.2. Steel Beam

### 3.2.1. Failure Process

The six steel frame beam specimens experienced a similar failure process, namely, the yielding stage, the elastoplastic stage, and the plastic failure stage under low cyclic repeated loading. The specimens were in the elastic stage at the initial stage of loading. With the increase of the displacement amplitude and cycle number, the specimens were in the elastoplastic stage. The local buckling was discovered at the bottom flange of the specimens. The local buckling deformation occurred with the web. With the further increase of the displacement amplitude and cycle number, the plastic hinge formed at the root of the beam, and the bearing capacity was reduced until specimen failure. However, the entire failure process of the specimens, which belongs to the ductile failure, is relatively slow and exhibits a good energy dissipation performance. Compared with uncorroded specimens, with the increase of the corrosion degree, the declines amplitude of the maximum flexural capacity of corroded specimens increases.

The maximum flexural capacity of B-2~B-6 specimens were obtained from Figure 8, as 128.35 kN, 123.81 kN, 110.68 kN, 102.07 kN, and 95 kN, respectively. By comparison with the B-1 specimen (136.9 kN) the decreased rates are calculated as 6.22%, 9.56%, 19.15%, 25.44% and 30.31%, respectively. At the same time, due to the buckling of the specimen bottom end flange, the web and the plastic hinge occurred, the lateral resisting stiffness of beam specimens was reduced. The typical failure model of the specimen is shown in Figure 9.

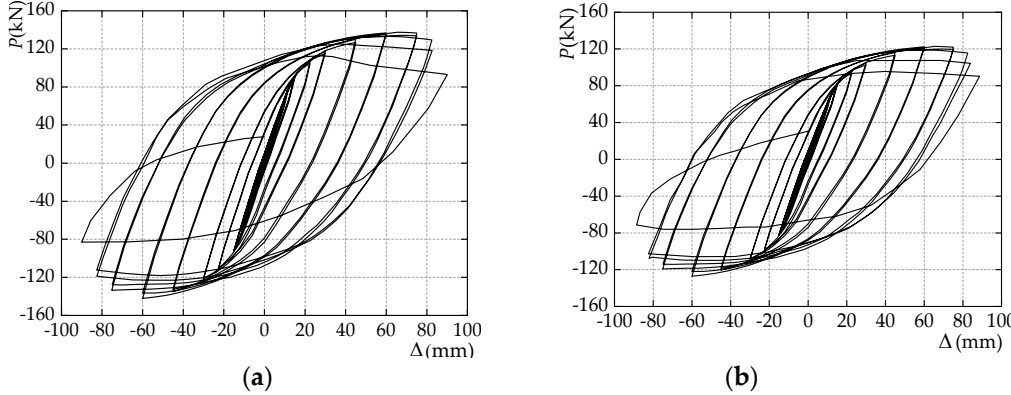

(a)                      (b)

**Figure 8.** *Cont.*

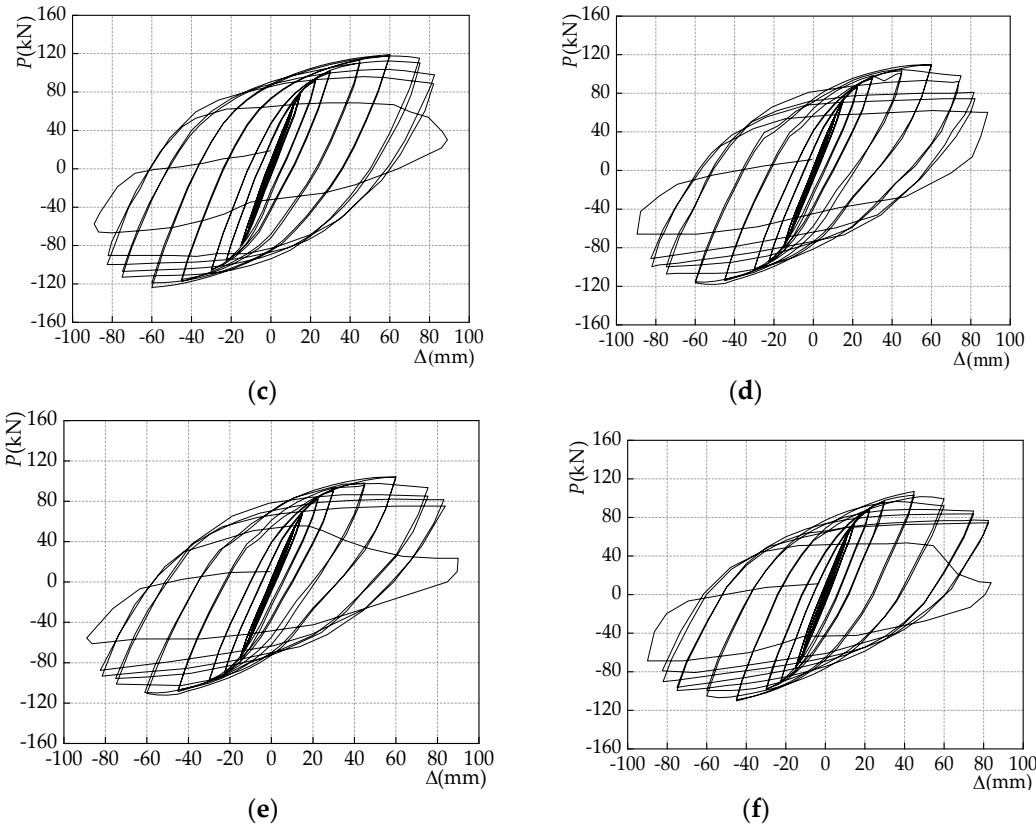

**Figure 8.** Loading-displacement curves of the specimens. (**a**) B-1; (**b**) B-2; (**c**) B-3; (**d**) B-4; (**e**) B-5; (**f**) B-6.

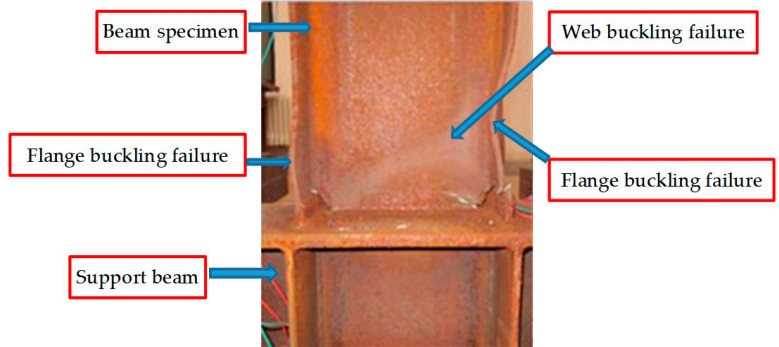

**Figure 9.** Typical failure model of beam bottom.

### 3.2.2. Hysteresis and Skeleton Curves

In this paper, the effect of different degrees of corrosion on the seismic performance of steel frame beams is considered. The low cyclic repeat loading test of steel frame beams is carried out, and the load (P) - displacement (Δ) hysteresis curves of the specimens are obtained, as shown in Figure 8. As can be seen:

1.  All of the load-displacement hysteresis curves, which show a distinct fusiform shape, are relatively full without pinching. The area surrounded by the hysteresis loops is relatively large, which means that the steel beam has good energy consumption and ductility.

2.  The conclusion is obtained by comparing the hysteresis curve of six specimens: Before the frame beam specimens reached the yield load, the slope of the loading curve changed little, the stiffness of specimen was basically unchanged, and it was in the elastic stage. As the displacement amplitude and the number of cycles increased, the specimens began to yield and the plastic deformation further increased, while the strength and stiffness deteriorated

significantly. Under the large displacement amplitude, the repeated loading caused the stiffness to degrade significantly. In general, the longer the corrosion time, the more obvious the stiffness and strength degradation.

The skeleton curves play a very important role in the elastoplastic analysis, which can reflect the yield displacement and load, peak displacement and load, ultimate displacement and capacity of the specimen. The skeleton curves of different degrees of corrosion are obtained by connecting the peak points of all the loops of the hysteresis curve in Figure 9 (starting at the unloading point), as shown in Figure 10.

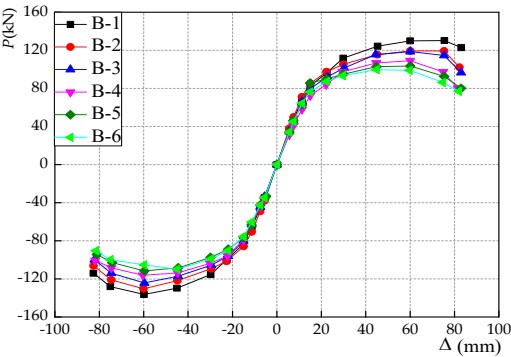

**Figure 10.** Skeleton curve of specimens.

As can be seen:

1.  Based on the above analysis, the failure process of the steel beam specimens is divided into the elastic stage, the elastoplastic stage, and the plastic failure stage. At the initial stage of loading, the skeleton curves developed linearly, the beam specimens were in the elastic stage. With the increase of displacement loading continually, skeleton curves appeared at an inflection point which is the yield point, indicating that the specimen reached the yield. As the loading displacement increased, the bearing capacity showed a nonlinear growth trend, and the lateral stiffness of the specimen decreased, the specimens were in the elastoplastic stage. As the loading displacement increased further, the skeleton curve began to decline after reaching the horizontal peak load, until the specimen was destroyed.
2.  The skeleton curve of the specimens with different degrees of corrosion basically overlapped at the initial stage of loading. However, with the increase of the continual loading displacement, the skeleton curve began to appear different. The greater the degree of corrosion, the more obvious the difference, such as the most severely corroded B-6 specimen, which were the most degraded.

## 4. Development of Restoring Force Model

### 4.1. Defining Skeleton Curve

#### 4.1.1. Simplified Skeleton Curve and Feature Points

Based on the above test results and analysis, a trilinear skeleton curve model is used in this paper, and the positive and negative skeleton curves are assumed to be symmetric, as shown in Figure 11. The points A (the yield point), B (the peak point), and C (the limit point) will be calculated later in this section, and then the skeleton curve model can be determined.

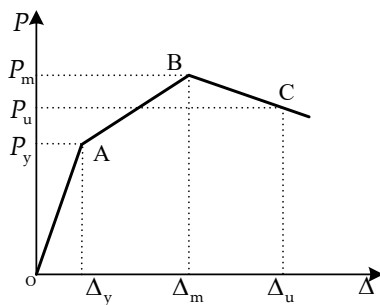

**Figure 11.** Simplified skeleton curve.

The specific calculation methods for each feature point of the corroded member are as follows:
(1) Yield point A
The yield load $P_y$ of the specimen is:

$$P_y = \frac{M_e}{L} = \frac{W_n}{L} f_y \tag{5}$$

where $M_e$ is the bottom moment of the beam end when calculating the elastic limit, $W_n$ is the section modulus of H-section steel beam, $L$ is the length of steel beam, and $f_y$ is the measured yield strength of steel.

Considering the influence of corrosion damage, Equation (2) is substituted into Equation (5) to obtain the yield load of the steel beam $P_y'$ as follows:

$$P_y' = \frac{W_n}{L} f_y' = \frac{W_n}{L} (1 - 0.9721 D_w) f_y \tag{6}$$

In a similar way, based on the theory of material mechanics, the yield displacement of the corroded beam $\Delta_y'$ is derived as follows:

$$\Delta_y' = \lambda \frac{P_y' L^3}{3 E_s' I} = \lambda \frac{(1 - 0.9721 D_w) f_y L^3}{3(1 - 0.9019 D_w) E_s I} \tag{7}$$

where $I$ is the section moment of inertia, $E_s$ is the elastic modulus of the steel material, and $\lambda$ is the correction factor that mainly considers the stiffness decrease of the support beam caused by the deformation or sliding and results in the actual displacement of the test specimen when the yielding is significantly larger than the theoretical displacement.

(2) Peak point B
Due to the existence of the hardening stage of the steel beam, the flat section assumption is no longer true, and it is difficult to derive the peak point bending moment $M_{max}$ through a theory that would exceed the plastic limit moment of the beam end $M_p$. Therefore, in the combined test results, the $M_{max}$ can be taken as:

$$M_{max} = 1.2 M_p \tag{8}$$

where,

$$M_p = W_p f_y \tag{9}$$

$$W_p = B t_f (H - t_f) + \frac{1}{4} (H - 2t_f)^2 t_w \tag{10}$$

where $W_p$ is the plastic section modulus of the H-shaped steel beam, $B$, $H$ are the section width and height of H steel, respectively, $t_f$ is the flange plate thickness, and $t_w$ is the web thickness, the rest of the parameter definitions are the same as before. The peak load $P_{max}$ of the beam specimen is:

$$P_{\max} = \frac{M_{\max}}{L} \tag{11}$$

Considering the influence of corrosion damage, Equation (2) is substituted into Equation (11) to obtain,

$$P'_{\max} = \frac{M'_{\max}}{L} = 1.2W_{\mathrm{p}}f'_{\mathrm{y}} = 1.2W_{\mathrm{p}}(1 - 0.9721D_{\mathrm{w}})f_{\mathrm{y}} \tag{12}$$

where $M'_{\max}$, $P'_{\max}$ are the peak moment and the load after corrosion, respectively.

Statistical analysis was performed on the test results, and the expressions of peak displacement $\Delta'_{\mathrm{m}}$ is obtained by the regression analysis considering the influence of corrosion:

$$\Delta'_{\mathrm{m}}/\Delta_{\mathrm{y}} = 3.565 - 0.00165e^{53.98D_{\mathrm{w}}} \tag{13}$$

(3) Ultimate point C

The ultimate load can take 85% of the peak load, namely:

$$P'_{\mathrm{u}} = 0.85P'_{\max} \tag{14}$$

where $P'_{\mathrm{u}}$ is the ultimate load of the beam when considering the influence of the corrosion damage.

The relationship between the ratio of the ultimate displacement and the yield displacement as a function of the weight loss rate is as follows:

$$\Delta'_{\mathrm{u}}/\Delta_{\mathrm{y}} = 4.634 + 0.219e^{-6.69D_{\mathrm{w}}} \tag{15}$$

where $\Delta'_{\mathrm{u}}$ is the ultimate displacement when considering the influence of the corrosion.

### 4.1.2. Verifying Skeleton Curves

Figure 12 shows the results of the comparison with the calculation value and test value of the beam skeleton curve. It shows that the calculation value obtained by the above method for determining the feature point is in good agreement with the test value, indicating that the skeleton curve obtained by the above method has better accuracy. Therefore, the above formula can be used as a method for determining the skeleton curve of steel beam members when considering the influence of corrosion.

### *4.2. Defining Hysteretic Rule*

Under the cyclic repeat loading, the structure components produce a certain degree of damage after yielding, and the damage accumulates, which leads to the continuous deterioration of the stiffness and strength of the component. At the same time, the more serious the corrosion of the component, the more serious the accumulation damage. Therefore, based on the Ibarra-Krawinkler (IK) model, the hysteresis rule of steel frame beams is established by introducing the cyclic degradation index, which considers different corrosion degrees, to objectively reflect the degradation law of corroded component strength and stiffness in an acidic environment.

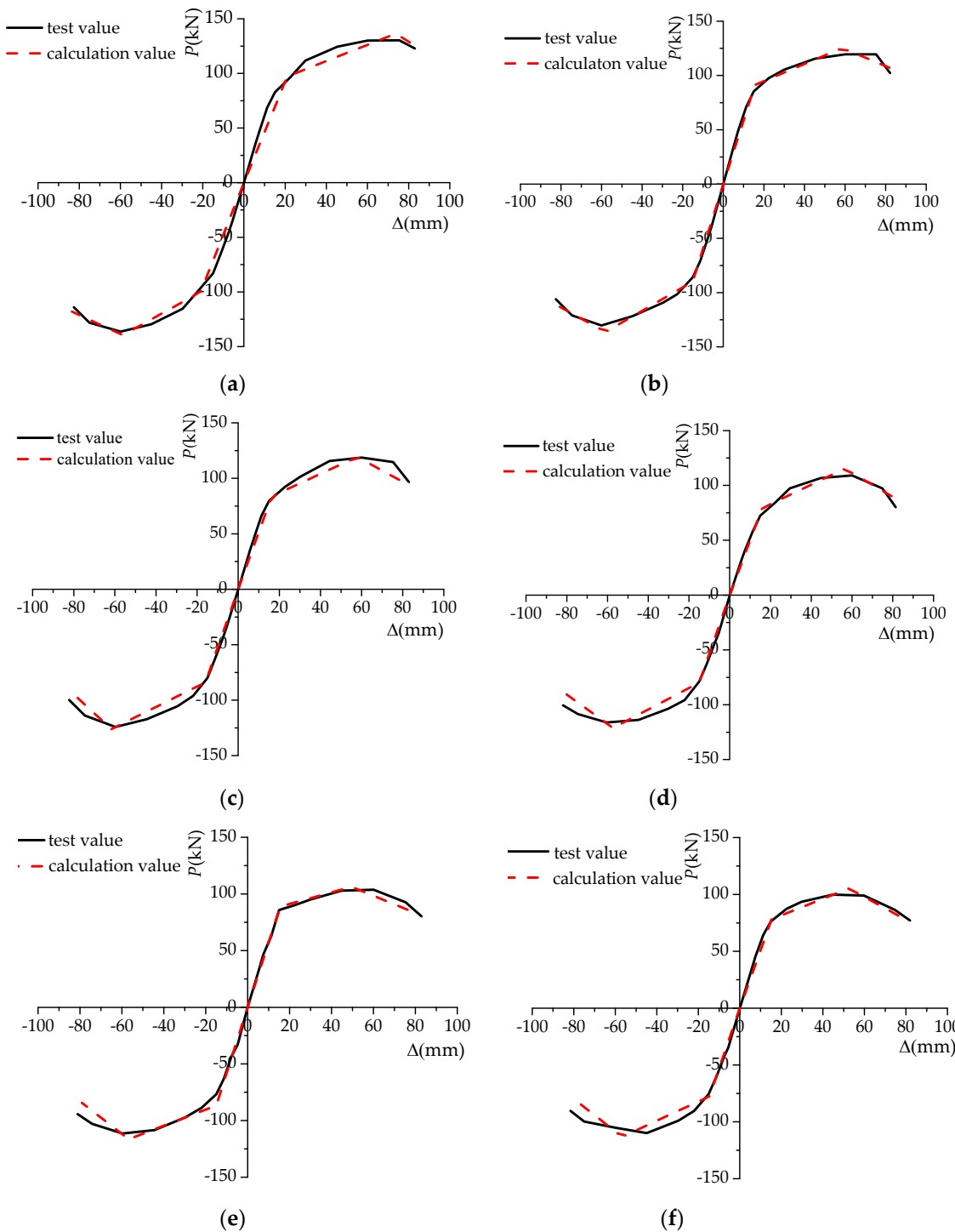

**Figure 12.** Comparing the calculation value with the test value. (**a**) B-1; (**b**) B-2; (**c**) B-3; (**d**) B-4; (**e**) B-5; (**f**) B-6.

### 4.2.1. Rule of Strength and Stiffness Degradation

(1) Cycle degradation index

The cyclic degradation index was used to reflect the cyclic degradation effect of the mechanical properties of corroded steel frame beam specimens in this paper [41], namely:

$$\beta_i = [E_i/(E_t - \sum_{j=1}^{i} E_j)]^c \tag{16}$$

where,

$$E_t = \Lambda \cdot M_y \tag{17}$$

$$\Lambda = 495 \cdot \left(\frac{h}{t_w}\right)^{-1.34} \cdot \left(\frac{b_f}{2 \cdot t_f}\right)^{-0.595} \cdot \left(\frac{c_{unit}^2 \cdot f_y}{355}\right)^{-0.360} \tag{18}$$

where $c$ is the rate of the cyclic degradation, $1 \le c \le 2$, in this paper it is equal to 1.5, $E_i$ is the energy dissipated by the specimen during the ith cycle loading, $\sum_{j=1}^{i} E_j$ is the cumulative dissipative energy of the specimen before the ith cycle loading, $E_t$ is the ability of energy dissipation of specimen, $\Lambda$ is the cumulative plastic angles of beam specimens, $M_y$ is the effective yield moment, in this paper, it is equal to $1.17M_p$, $h$ is the web height, $b_f$ is the flange plate width, $c_{unit}^2$ is the unit conversion factor, which is equal to 1 when using millimeters and megapascals, and the rest of the parameter definitions are the same as before.

Considering the influence of corrosion damage, the cumulative plastic angles of beam specimens $\Lambda$ can be rewritten by combining Equation (2):

$$\Lambda = 495 \cdot \left(\frac{h}{t_w}\right)^{-1.34} \cdot \left(\frac{b_f}{2 \cdot t_f}\right)^{-0.595} \cdot \left[\frac{c_{unit}^2 \cdot (1 - 0.9721 D_w) f_y}{355}\right]^{-0.360} \tag{19}$$

(2) Strength degradation rules

According to the test results, the yield load decreases with the increase of the displacement amplitude and the number of cycles, and the peak load of the hysteresis loop decreases with the increase of the number of cycles under the same displacement amplitude after the specimen enters the elastoplastic stage. The strength degradation rules can be defined as:

$$P_{y,i}^{\pm} = (1 - \beta_i) P_{y,i-1}^{\pm} \tag{20}$$

$$P_{j,i}^{\pm} = (1 - \beta_i) P_{j,i-1}^{\pm} \tag{21}$$

where, $P_{y,i}^{\pm}$, $P_{y,i-1}^{\pm}$ are the yield load of the specimen at the time of the ith and i-1th cyclic loading respectively, $P_{j,i}^{\pm}$, $P_{j,i-1}^{\pm}$ are the peak load of the specimen under the ith and i-1th cycle loading at jth stage displacement respectively. The superscript "$\pm$" indicates the loading direction, where "+" is positive loading and "$-$" indicates reverse loading. The specific strength degradation rules are shown in Figure 13.

(3) Stiffness degradation rules

The unloading stiffness of the specimen is also degraded continuously under the action of repeat cyclic loading, which can be described by the following formula:

$$K_{u,i} = (1 - \beta_i) K_{u,i-1} \tag{22}$$

where $K_{u,i}$, $K_{u,i-1}$ are the unloading stiffness of specimen during the ith and i-1th cycle loading, respectively.

In addition, the hysteresis curves obtained from the test shows that the reloading stiffness is degraded since the last cycle unloading stiffness, and the degradation law of the reloading stiffness can be characterized as:

$$K_{r,i} = (1 - \beta_i) K_{u,i-1} \tag{23}$$

where $K_{r,i}$ is the reloading stiffness of specimen during the ith cycle loading. The specific stiffness degradation rules are shown in Figure 14.

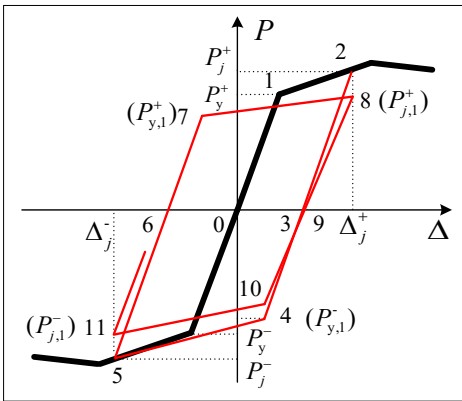

**Figure 13.** Strength degradation rules.

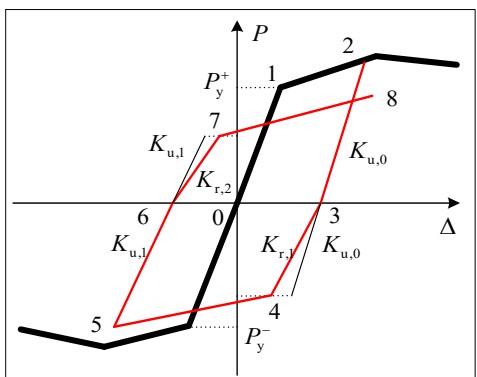

**Figure 14.** Stiffness degradation rules.

In summary, according to the established skeleton curves and hysteresis rules, the restoring force model of corroded steel frame beams in the acidic atmospheric environment is shown in Figure 15. The specific calculation steps are obtained as follows:

Step 1: The feature points of the skeleton curve for the steel beam with different degrees of corrosion were calculated according to the Equations (5)–(15).

Step 2: A skeleton curve based on the assumption of the center-symmetry of the positive and negative directions was drawn.

Step 3: The loading and unloading in the forward and reverse directions along the skeleton curve, where the loading and unloading paths coincide before the yield (0–1 segment) was performed.

Step 4: From yielded to peak load, the loading path was along the skeleton curve (1–2 segment), the unloading stiffness was taken as the initial elastic stiffness $K_e$, namely, $K_e = P'_y / \Delta'_y$, the reverse loading was continually carried to the negative yielding point 4, and the yield loading and the loading stiffness were determined by Equation (20) and Equation (23) respectively. The unloading after loading to the negative direction target point 5 on the skeleton curve, at this time the unloading stiffness was determined according to Equation (23). After the positive reloading (6–7 segment), the yield load and reloading stiffness were re-determined again by Equation (20) and Equation (23) respectively. When the second cyclic loading at the same displacement amplitude was carried out, the positive peak loading point of the hysteresis loop degenerated to point 8, and the load value was calculated according to Equation (21), but the displacement value was unchanged. The unloading stiffness was re-determined according to Equation (22) when unloading from point 8. After continuing with the reverse loading, the negative yield loading and reloading stiffness were re-determined by Equations (20) and (23). The negative peak loading point of the hysteresis loop degenerated to point 11, and the load value was calculated according to Formula (21), but the displacement value was unchanged. When unloading again, the unloading stiffness was re-determined according to Equation (22).

Step 5: Except for the "softening section" of the hysteresis loop, the loading path calculation method for each displacement amplitude after the peak load was similar to step 6. For example, the peak loading point of the positive skeleton curve degraded to point 14, which can be calculated through Equation (21), but the target was point 15, which is on the skeleton curve, and the load value was equal to the load value of point 14.

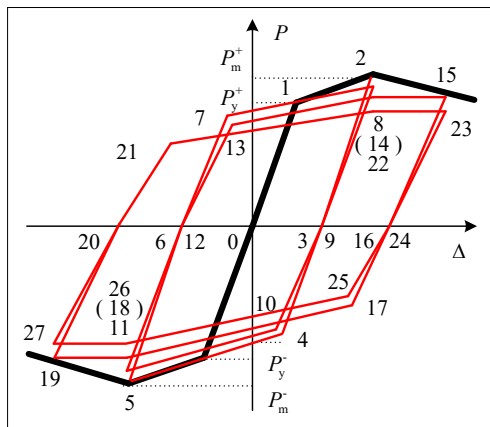

**Figure 15.** Restoring force model of corroded steel frame beam.

### 4.2.2. Comparative Study of Hysteresis Performances

In order to verify the correctness of the restoring force model established in this paper, the hysteresis curve calculated according to the above method was compared with the hysteresis curve obtained by the experiment, and the results are shown in Figure 16, where the data of the specimen DB700 and TRS2A are used from Reference [42] and [43].

The results show that the calculation hysteresis curves well agree with the experimental hysteresis curves, which can better reflect strength and stiffness degradation. The restoring force model established in this paper can accurately reflect the seismic performance of corroded steel frame beams.

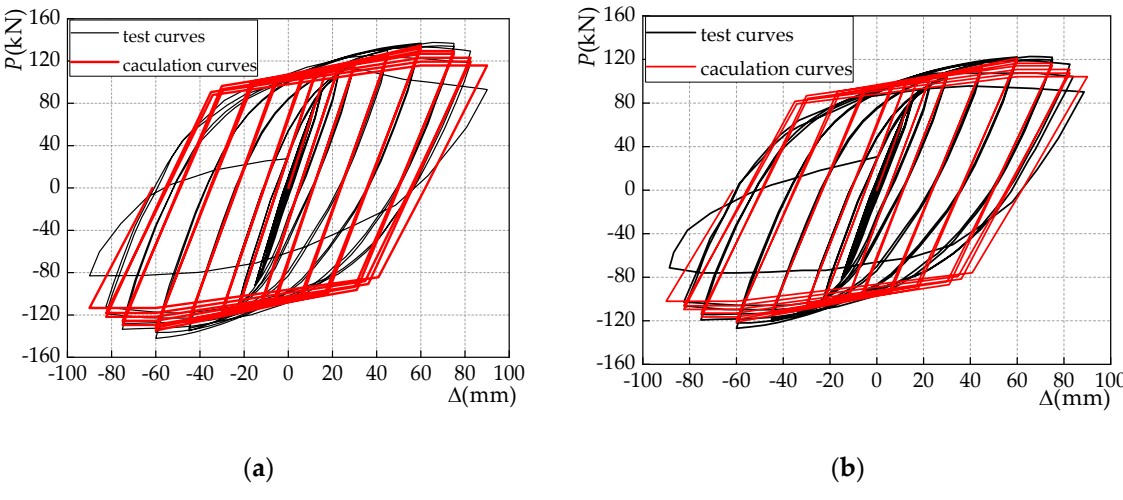

(**a**)                          (**b**)

**Figure 16.** *Cont.*

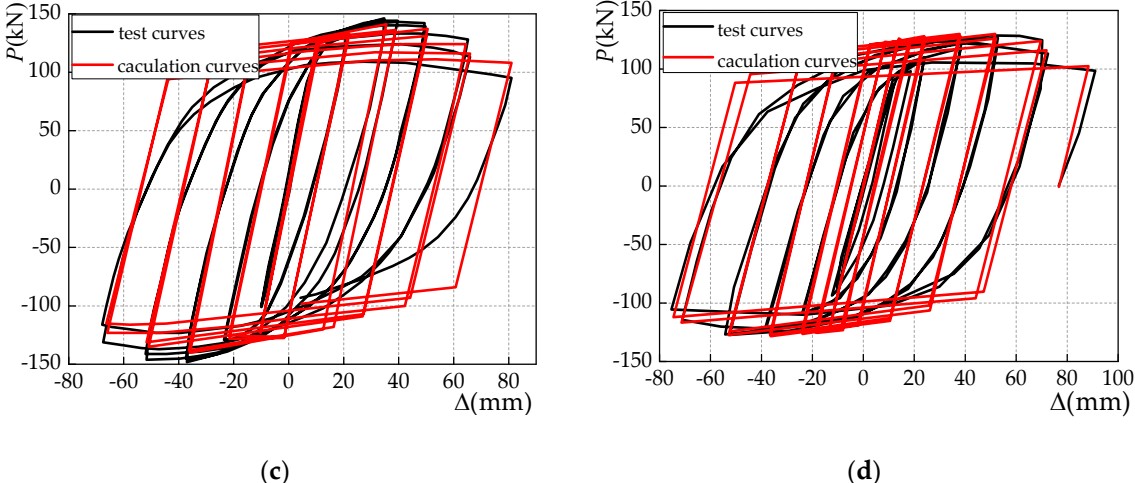

**Figure 16.** Comparison of calculation hysteresis curves and test hysteresis curves. (**a**) Specimen B-1; (**b**) specimen B-2; (**c**) specimen DB700; (**d**) specimen TRS2A.

## 5. Conclusions

In this paper, a new restoring force model for the simulation of the corroded steel frame beam under horizontal cyclic loading was developed, and some conclusions were obtained as follows:

1.  The test results of the corroded steel material verified a widely known fact that corrosion has a significant effect on the mechanical properties of steel. With the increase of the corrosion degree, the mechanical properties of steel have a linear decreasing trend, and the linear regression analysis is carried out to obtain the expressions of various mechanical properties and the weight loss rate.

2.  The test results of the corroded steel frame beam under the cyclic loading show that the failure process of the steel frame beam specimen experiences three stages, namely, elasticity, elastoplasticity, and the plastic failure stage. The bottom flange plate is partially buckled first, then the web is convexly curved, and finally the plastic hinge is formed at the bottom of the beam. However, with the increase of the corrosion degree, the horizontal bearing capacity of the specimens is gradually reduced, and the stiffness is significantly degraded. The displacements corresponding to the end flange buckling, the web drum, and the plastic hinge formation are also gradually reduced.

3.  Based on the above results, the skeleton curve of the corroded steel beam specimen is simplified to a trilinear model considering the descent segment. The theoretical derivation formulas of the three characteristic points are given in the model. There is good agreement between the calculation value and test value, demonstrating that the established trilinear model in this paper is reasonable.

4.  The cyclic degradation index is introduced by considering the effect of the corrosion degree for the hysteresis performance, and the restoring force model of the corroded steel frame beam is established by defining the degradation rule of strength and stiffness. The computed hysteresis curves are in good agreement with the experimental ones, verifying the applicability of the restoring force model. The developed model truly reflects the hysteretic behavior of the corroded steel frame beam, and the research results provide a theoretical basis for the nonlinear seismic response analysis of corroded steel frame structures.

5.  Considering the rapid development of structural health monitoring (SHM), we will conduct some investigation on the corrosion monitoring of steel frame beams in the future, including the active sensing method [44–46], and the electro-mechanical impedance method [47].

**Author Contributions:** B.W. and W.H. designed and performed the experiments; B.W. and S.Z. analyzed the data; B.W. and W.H. wrote the paper.

**Acknowledgments:** The authors appreciate the support of the National Science and Technology Support Program (2013BAJ08B03), the National Science Foundation of China (51678475), the Natural Science Foundation of Shaanxi Province (2018JQ5158) and the Scientific Research Program Funded by Shaanxi Provincial Education Department (18JK0382).

**Conflicts of Interest:** The authors declare no conflict of interest.

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
