# Peer review of "Study on Restoring Force Performance of Corrosion Damage Steel Frame Beams under Acid Atmosphere"

_applsci, doi:10.3390/app9010103_

Round 1
Reviewer 1 Report
· There is room for improvement in terms of English writing and editorial revisions. Below are some examples:
- Line 31: “such as poor of fire resistance”. It is not correct. It should be “poor fire resistance”.
- Line 124: “following the currently Chinese national code [47-49]”.Revise.
- Line 396: “a new the restoring force model for the….”. Revise
- In figure 9, 10, 12 and 16 “P/kN” is not the correct way to label the axes. It should be P (kN). Similar comment for the horizontal axis which is the displacement.
As I said, there are many other locations that need to be improved. These are just some samples.
· The Introduction should be re-written since it is not introducing the problem investigated in an appropriate way. Below are some examples:
- Line 29: What do you mean by “good connection”.Any type of connection can be good or bad depending whether it is designed correctly or not. Rephrase.
- Line 32: Fatigue cracks in steel beams due to wind load is not common at all; As we are not designing the regular steel connection in the building for fatigue unless it is subjected to repeated loads e.g. crane loading.
- Having one sentence referring to several references (“[8-13] and [16-22]) deemed to be superficially just touch on the surface although it is expected to go in depth in a technical manuscript. I suggest to reduce the number of references and consider including the ones which are directly related to the topic of the current manuscript with providing more details.
- Line 73: “According to above analysis”; no analysis has been done above. Revise.
- There are too many unnecessary references in the introduction that should be deleted. However, a section needs to add on previous research on steel members in a harsh corrosive environment. I cannot see any explanation about the corrosion of steel members. The researches related to the behaviour of steel members as well as structures in the coastal areas might be helpful.
· Table 1 is not clear at all. Quantity “16”: is it for all the thicknesses? How many replicas for each type? If yes, you don’t need a table for saying this. Revise.
· Line 134: It in not an instability necessarily. The specimen might slightly deflect in the out-of-plane direction. Revise.
· Line 135: AISC 341 is Called “Seismic Provisions for Structural Steel Buildings” not "Steel Structure Seismic Code". Also, the loading regime in Figure 6 is not from AISC41, to the best of my knowledge. Please revise.
· The conclusion mentioned in line 148 to 150 is obvious even without performing the tests. It needs to be re-worded to provide more informative statement for the readers.
· In figure 7, the “x” and “y” needs to be changed to correct parameters.
· Line 178 to 180: What is the value for the bearing capacity? How did you calculate it? The specimen is failing due to moment. So the flexural capacity of the section in more of interest rather than bearing. Please clarify.
· Line 181 to 184: Please mention the original values so the reader can sense the reduction percentages.
· Figure 8: I believe the failure modes is very much dependant on the connection type connecting the “beam specimen” to the “support beam”. If you are interested to investigate the behaviour of steel sections, you should have made it independent of the connection types e.g. picking a simply supported beam and monitor the failure modes at the middle of the span where you had pure moment with minimum effect of connection types. What you have investigated here is not the steel beam but a full welded steel connections. In other words, the test setup considered is basically for investigating the steel connection behaviour not the steel sections.
· Line 194: “better” than what? What are you comparing with?
· I suggest to normalize the vertical and horizontal axes values by dividing the values by yielding force and displacement, so it can be independent of the section size.
· For the beams, it is suggest to report the results in terms of moment not the concentrated load at the tip of cantilever. In other words, the regression analysis should be exercised on the normalized flexural curves to be more versatile.
Author Response
Point 1: There is room for improvement in terms of English writing and editorial revisions. Below are some examples:
- Line 31: “such as poor of fire resistance”. It is not correct. It should be “poor fire resistance”.
- Line 124: “following the currently Chinese national code [47-49]”.Revise.
- Line 396: “a new the restoring force model for the….”. Revise
- In figure 9, 10, 12 and 16 “P/kN” is not the correct way to label the axes. It should be P (kN). Similar comment for the horizontal axis which is the displacement.
Response 1: The comment is agreed. We have proofread the manuscript carefully, and the writing has been improved. Some examples are given as follows,
Line 31: the “poor of fire resistance” has been amended as “poor fire resistance”.
Line 124: “following the currently Chinese national code [47-49]” has been revised as “following the current Chinese standard [47-49]”.
Line 396: “a new the restoring force model for the….” has been revised as “a new restoring force model for the …”
The unites of abscissa and ordinate in Figure 9, 10, 12, and 16 have been modified as comment.
Point 2: As I said, there are many other locations that need to be improved. These are just some samples.
- The Introduction should be re-written since it is not introducing the problem investigated in an appropriate way. Below are some examples:
- Line 29: What do you mean by “good connection”. Any type of connection can be good or bad depending whether it is designed correctly or not. Rephrase.
- Line 32: Fatigue cracks in steel beams due to wind load is not common at all; As we are not designing the regular steel connection in the building for fatigue unless it is subjected to repeated loads e.g. crane loading.
- Having one sentence referring to several references (“[8-13] and [16-22]) deemed to be superficially just touch on the surface although it is expected to go in depth in a technical manuscript. I suggest to reduce the number of references and consider including the ones which are directly related to the topic of the current manuscript with providing more details.
- Line 73: “According to above analysis”; no analysis has been done above. Revise.
- There are too many unnecessary references in the introduction that should be deleted. However, a section needs to add on previous research on steel members in a harsh corrosive environment. I cannot see any explanation about the corrosion of steel members. The researches related to the behaviour of steel members as well as structures in the coastal areas might be helpful.
Response 2: The comment is appreciated. The introduction has been improved significantly as follows partially,
- The “good connection” has been amended as “easy-to-connection”.
- The “fatigue cracks under wind” is deleted.
- The references that are related to the topic of this paper directly have been explained in detail. Additionally, the numbers of references in one sentence have been reduced.
- The “According to above analysis” in line 73 has been revised as “Based on the above-mentioned introduction”.
- The unnecessary references in this manuscript has been deleted, meanwhile, a section about investigation of steel members in corrosive environments has been added.
Point 3: Table 1 is not clear at all. Quantity “16”: is it for all the thicknesses? How many replicas for each type? If yes, you don’t need a table for saying this. Revise.
Response 3: The quantity “16” is for each type (repeat 2 times under 8 acceleration corrosion time). We have revised the table 1 to make it more clear.
Point 4: Line 134: It in not an instability necessarily. The specimen might slightly deflect in the out-of-plane direction. Revise.
Response 4: Though the deflection in the out-of-plane direction may be slight, we only used the lateral support to ensure the structural stable. However, we amended this sentence to clarify the issue under reviewer’s comment.
Point 5: Line 135: AISC 341 is Called “Seismic Provisions for Structural Steel Buildings” not "Steel Structure Seismic Code". Also, the loading regime in Figure 6 is not from AISC41, to the best of my knowledge. Please revise.
Response 5: The comment is agreed. Sorry for our carelessness. We have revised the name of AISC 341-10 as “Seismic Provisions for Structural Steel Buildings”. Additionally, sorry for our mistake, it is not from AISC 341-10. The new reference has been quoted in this manuscript.
Point 6: The conclusion mentioned in line 148 to 150 is obvious even without performing the tests. It needs to be re-worded to provide more informative statement for the readers.
Response 6: The comment is agreed. We have rewritten these sentences to provide more information.
Point 7: In figure 7, the “x” and “y” needs to be changed to correct parameters.
Response 7: The “x” and “y” in Figure 7 have been changed to correct parameters.
Point 8: Line 178 to 180: What is the value for the bearing capacity? How did you calculate it? The specimen is failing due to moment. So the flexural capacity of the section in more of interest rather than bearing. Please clarify.
Response 8: The comment is appreciated. Here, we just develop a qualitative analysis. Thus, we didn’t calculate the bearing capacity. Additionally, we have revised the bearing capacity as the flexural capacity to make the manuscript more clear.
Point 9: Line 181 to 184: Please mention the original values so the reader can sense the reduction percentages.
Response 9: The original values have been provided in this manuscript, to make readers more intuitive.
Point 10: Figure 8: I believe the failure modes is very much dependant on the connection type connecting the “beam specimen” to the “support beam”. If you are interested to investigate the behaviour of steel sections, you should have made it independent of the connection types e.g. picking a simply supported beam and monitor the failure modes at the middle of the span where you had pure moment with minimum effect of connection types. What you have investigated here is not the steel beam but a full welded steel connections. In other words, the test setup considered is basically for investigating the steel connection behaviour not the steel sections.
Response 10: The comment is considered. In this paper, as the title states that “steel frame beams”, we emphasis the restoring force performance of the frame. In other words, Figure 8 mainly represents the failure pattern of the welded frame beam under low-cycle repeated load. The support beam serves as the fixed end of the beam specimen, and the welding is used to simulate the rigid connection. Instead of the connection method test, the beam failure test conducted in this paper mimics the seismic performance. Thus, it is not necessary to monitor the moment at the middle of the span. The plastic hinge usually appears at the end of the frame beam under the earthquake. As depicted in Figure 8, the test results are consistent with the calculation and analysis results, and the failure phenomenon is obvious.
Point 11: Line 194: “better” than what? What are you comparing with?
Response 11: The comment is appreciated. Sorry for the typo. It should be “good”. We mean that the steel beam’s performances of energy consumption and ductility are good, considering the large areas surrounded by the hysteresis loops.
Point 12: I suggest to normalize the vertical and horizontal axes values by dividing the values by yielding force and displacement, so it can be independent of the section size.
Response 12: The comment is considered. However, considering the section size may has other effect on the seismic performance of structures, we still used the present figures, to ensure the accuracy of the numerical model and calculation results, and have a direct comparison with the experimental results.
Point 13: For the beams, it is suggest to report the results in terms of moment not the concentrated load at the tip of cantilever. In other words, the regression analysis should be exercised on the normalized flexural curves to be more versatile.
Response 13: The comment is considered. However, in fact, both moment- curvature and load- displacement are generally used to describe the seismic performance of frame beams, such as,
(1) Jin L.; Du, X.L.; Li D.; Su X.. Seismic behavior of RC cantilever beams under low cyclic loading and size effect on shear strength: An experimental characterization. Engineering Structures. 122, 2016, 93-107.
(2) Uchida R.; Hamahara M.; Suetsugu H.; SATO N.; OSAKI K.. Restoring force characteristics model of prestressed concrete beams in beam-column assemblies. J. Struct. Constr. Eng.. 69(575), 2004, 105-112.
(3) Yu, D.H.; Wang C.H.. Restoring force model of semi seam coupling beam based on ABAQUS. World Earthquake Engineering, 33(2), 2017,89-96·
Thus, in this paper, we still use the concentrated load - displacement curve to report the results.

Reviewer 2 Report
This paper is well written. The state of the art part is satisfactory. The presentation is technically correct. The experimental design is appropriate. The test results and theoretical formulas are discussed in detail. The established model for the simulation of the corroded steel frame beam under cyclic loading is reasonable. The conclusions are justified.
I think that the manuscript will be definitely worthy of publication after the requested minor improvements. In the attached document, you will find my detailed comments.

Author Response
Point 1: What do you mean with ‘and so on’ (line 63)?
Response 1: The comment is appreciated. The sentence has been revised as “The more classic PH model includes the bilinear and trilinear hysteresis model which consider several factors including the concrete cracking, yielding, loading and unloading stiffness degradation, etc.”
Point 2: I don’t understand this sentence. Please, clarify (line 87-89).
Response 2: The comment is agreed. Sorry for unclear. The sentence has been clarified as “To obtain the mechanical properties of the corroded steel such as the yield strength, the ultimate strength, the elongation, and elastic modulus, the tests were conducted based on standards. Then, above-mentioned fundamental parameters that were used in the seismic analysis of corroded frame beam were determined in this paper.”
Point 3: Please add all used test standards and design codes in the references (line 96).
Response 3: The comment is appreciated. All the used standards have been quoted in the references.
Point 4: Unlimited or ultimate strength (line 159).
Response 4: The comment is agreed. Sorry for typo. It should be “ultimate”.
Point 5: I don’t understand this sentence. Please, clarify (line 234-235).
Response 5: The comment is appreciate. Sorry for confusion. The sentence has been clarified as “The points A (the yield point), B (the peak point), and C (the limit point) will be calculated later in this section, and then the skeleton curve model can be determined.”.
Point 6: This statement is obvious (line 398-399).
Response 6: The comment is agreed. The sentence has been amended as “The test results of corroded steel material verified a widely known fact that the corrosion has a significant effect on the mechanical properties of the steel”.
Round 2
Reviewer 1 Report
No other comments...